# Adaptive Prescribed-time control of Dynamic Positioning ships based on Neural networks

1st Yongsheng Dou
*College of Navigation*
*Dalian Maritime University*
Dalian, China
dysheng@dlmu.edu.cn

2nd Chenfeng Huang*
*College of Navigation*
*Dalian Maritime University*
Dalian, China
chenfengh@dlmu.edu.cn

3rd Yi Zhao
*College of Navigation*
*Dalian Maritime University*
Dalian, China
yi_zhao@dlmu.edu.cn

*Abstract*—In this paper, a novel controller with prescribed-time performance is designed for dynamic positioning (DP) system of ships with model uncertainty and unknown time-varying disturbances. Initially, an error transformation function with zero initial value is introduced by constructing fixed-time funnel boundaries (FTFBs) and a fixed-time tracking performance function (FTTPF). The proposed controller ensures stable convergence of the new error, maintaining it within fixed upper and lower boundaries. When the prescribed time is reached, the system state will achieve prescribed-time (PT) stability. Secondly, by deploying radial basis function neural networks (RBF-NNs) and dynamic surface control (DSC), adaptive controller with simple forms are rationally applied to Backstepping technology, and the uncertain terms of the system are approximated online, the singularity and complexity explosion problems of the ship control system are also addressed. In addition to that, the stability analysis results of the system prove that all errors of the closed-loop system are semi-global uniformly ultimately bounded (SGUUB) stable. Finally, the simulation results on a DP ship confirm the superiority of the proposed scheme.

*Index Terms*—Dynamically positioned ships, prescribed-time control, fixed-time funnel boundaries, Backstepping

## I. INTRODUCTION

In marine engineering, dynamic positioning (DP) systems are critical for maintaining precise ship positions and orientations in the marine environment [1]. These systems enable ships to hold exact positions or follow predetermined paths without anchoring by utilizing thrusters and power systems. This capability is crucial for marine engineering operations, including oil and gas drilling, underwater pipeline installation, and cable laying. As marine operations grow more complex, traditional DP methods encounter significant challenges, such as environmental disturbances, system parameter uncertainties, and operational efficiency concerns [2].

Consequently, researchers are increasingly adopting advanced control strategies to improve the performance and adaptability of DP systems. However, the highly nonlinear terms of ship dynamics and the continuously changing marine environment often cause traditional control methods to struggle under extreme conditions. Furthermore, most existing

DP control strategies depend on extended control processes to achieve stability [3], which may not always be the optimal solution. Therefore, developing a control strategy that can respond quickly and complete tasks within a prescribed time is particularly crucial.

With the rapid advancement of control technologies and methods in recent years, DP systems have been more broader application in maritime operations and offshore exploration for ships and drilling platforms. For instance, in the presence of unknown ship parameters, [4] developed a robust adaptive observer for DP systems, capable of estimating ship velocities and unknown parameters under external disturbances. An adaptive observer based on neural networks (NNs) was developed to estimate the velocity data of the unmanned surface vessel (USV) in [5], even though both the system parameters and nonlinearities of the USV were presumed to be uncertain. NNs approximation techniques are used to compensate for uncertainty and unknown external disturbances, removing the prerequisite for a priori knowledge of ship parameters and external disturbances. Meanwhile, MLP technology is employed to address the computational explosion problem [6] [8]. However, in [7], static NNs are used for control force and moment allocation of an over-actuated ship by measuring the thruster force and commands and gathering data for practice of the NNs.

Due to the time-varying boundary functions can achieve prescribed performance of dynamic system on both transient and steady-state phases, [10] proposed a novel boundary function control approach and introduced an error transformation function, showing training stability of the closed-loop systems with prescribed transient and steady-state functions. In the field of marine engineering operations, [11] proposed a robust adaptive prescribed performance control (RAPPC) law by constructing a concise error mapping function and achieved the DP prescribed performance control. To address positioning error constraints, input saturation and unknown external disturbances, [12] proposed a variable gain prescribed performance control law and constructed the error mapping functions to integrate the prescribed performance boundary to the controller design. Soon after that, a robust fault-tolerant control allocation scheme is developed to distribute again the forces among faulty actuators in [13]. Its performance function

This work is partially supported by the Fundamental Research Funds for the Central Universities (Grant No.3132024129), the Fisrst-Class Disciplines Cross Research Project (Grant No.2023JXA04). (Corresponding author: Chenfeng Huang.)

is united with an auxiliary in-between control technique to create a high-level controller.

Inspired by the above research work, The contributions of this paper are as follows:

1) Building upon the research foundation of reference [11], this article proposes an adaptive prescribed-time control scheme for DP system of ship with model uncertain and unknown environment disturbances. Unlike the reliance on initial conditions discussed in reference [10], the construction of the fixed-time tracking performance function (FTTPF) ensures that the controller's prescribed performance is no longer dependent on initial conditions. Furthermore, the new dynamic errors will deviate from an initial value of 0, remaining consistently confined within the set fixed-time tracking performance function (FTFBs).

2) Based on NNs, unknown functions of the new dynamic error derivative terms and unknown model parameters of the ship are approximated online. In addition, the adaptive parameters based on weight allocation are reduced to two to compensate for the unknown gain function. The dynamic surface control (DSC) filtering technique is introduced to address the complexity explosion problem caused by the differentiation of the virtual controller, thereby reducing the computational burden. Finally, two comparative simulations of a DP ship is executed to demonstrate the effectiveness of the proposed algorithm.

## II. MATHEMATICAL MODEL OF DYNAMICALLY POSITIONED SHIPS AND PROBLEM FORMULATION

In the design of DP systems, a ship is considered a multi-input multi-output (MIMO) control system that includes dynamics influenced by mass, damping, stiffness, and external disturbances. On the basis of the seakeeping and maneuvering theory, the following three DOF nonlinear mathematical model is used to describe the dynamic behavior of the ship in the presence of disturbances [14]:

$$\dot{\eta} = J(\psi)v \tag{1}$$

$$M\dot{v} + D(v)v = \tau + \tau_d \tag{2}$$

where $\eta = [x, y, \psi]^\top \in \mathcal{R}^3$ represent the attitude vector including the surge position $x$, the sway position y and the heading $\psi \in [0, 2\pi]$ in the earth-fixed coordinate system. $v = [u, v, r]^\top \in \mathcal{R}^3$ denotes the velocity vector of the ship in the body-fixed coordinate system, which composed of the surge velocity $u$, sway velocity $v$ and yaw velocity $r$, respectively. $J(\psi)$ is the velocity transformation matrix as follow:

$$J(\psi) = \begin{bmatrix} \cos(\psi) & -\sin(\psi) & 0 \\ \sin(\psi) & \cos(\psi) & 0 \\ 0 & 0 & 1 \end{bmatrix} \tag{3}$$

with $J^{-1}(\psi) = J^\top(\psi)$ and $\|J(\psi)\| = 1$. Equation (4) gives the specific expression of the positive definite symmetric inertia matrix $M \in \mathcal{R}^{3 \times 3}$, which including additional mass.

Equation (5) gives the specific expression of the nonlinear hydrodynamic function $D(v)v$.

$$M = \begin{bmatrix} m - X_{\dot{u}} & 0 & 0 \\ 0 & m - Y_{\dot{v}} & mx_G - X_{\dot{r}} \\ 0 & mx_G - X_{\dot{r}} & I_z - N_{\dot{r}} \end{bmatrix} \tag{4}$$

$$D(v)v = \begin{bmatrix} D_1 \\ D_1 \\ D_3 \end{bmatrix}$$

$$D_1 = -X_u u - X_{|u|u}|u|u + Y_{\dot{v}}v|r| + Y_{\dot{r}}rr \tag{5}$$

$$D_2 = -X_{\dot{u}}ur - Y_v v - Y_r r - X_{|v|v}|v|v - X_{|v|r}|v|r$$

$$D_3 = (X_{\dot{u}} - Y_{\dot{v}}) uv - Y_{\dot{r}}ur - N_v v - N_r r - N_{|v|v}|v|v$$
$$\quad - N_{|v|r}|v|r$$

where $m$ are ship's mass, $I_z$ are moment of inertia and $X_u$, $X_{|u|u}$, $Y_{\dot{v}}$, etc., are every hydrodynamic force derivatives. It is obvious from the expression in (5) that the nonlinear damping force composed of linear and quadratic terms. In the controller design of this paper, $D(v)v$ is an uncertain term in which the structure and parameters are unknown and is approximated online using NNs in later section.

$\tau = [\tau_u, \tau_v, \tau_r]^\top \in \mathcal{R}^3$ denotes the control inputs, which are the forces and moments generated by the equipped actuators on the ship consisting of the shaft thruster, the tunnel thruster, and the azimuth thruster. In order to simplify the control inputs, all actuator devices inputs are fused into three degrees of freedom : $\tau_u$ in surge, $\tau_v$ in sway and $\tau_r$ in yaw. $\tau_d = [\tau_{du}, \tau_{dv}, \tau_{dr}]^\top$ indicates the unknown time-varying environment distraction induced by wind, and waves.

**Assumption 1.** *The environment disturbance $\tau_{dv}$ is bounded in the marine environment, indicating the existence of bounded $\bar{\tau}_{dv} > 0$ for $\tau_{dv}$. i.e., $|\tau_{dv}| < \bar{\tau}_{dv}$.*

**Remark 1.** *When modeling ship DP systems, it is often necessary to accurately characterize and predict the effects of environmental disturbances on the ship. In order to simplify the model and to facilitate the design and testing of control algorithms, these environmental disturbances can be approximated and modeled using a sine-cosine function. The frequency, amplitude and phase of the interference can be easily adjusted using the sine-cosine function to simulate different intensities and types of environmental conditions.*

In the setting of unknown time-varying disturbances and model uncertainty, the goal of the control is to find a control laws $\tau$ makes the ship's position $(x, y)$ and heading $\psi$ successfully reach the desired position $\eta_d$ within the prescribed time. At the same time, the constructed zero-initial-value error function also converges within the set boundaries within the settling time and arbitrarily small errors, and all the errors are bounded all the time.

## III. FUNNEL CONTROL AND FUNNEL VARIABLE

In the context of advanced control strategies for DP systems, particularly those addressing strict timing requirements, the concepts of FTFBs and FTTPF are integral. These are designed

to ensure that the control system adheres to performance metrics strictly within a settling interval, regardless of initial conditions. In this section, the definitions of FTFBs and FTTPF are introduced for the purpose of imposing error bounds on them and constructing new error functions.

## A. The Design Of Prescribed-time Funnel Boundary

**Definiion 1.** *[15] FTFBs define the permissible bounds within which the system's states must remain over time. These boundaries are set to compact over a fixed-time period, ensuring that the system's behavior converges to the desired state within the settling duration. These boundaries are particularly useful in scenarios where rapid and reliable system stabilization is crucial.*

*Equation (6) is selected as an FTFBs with the following traits: (1) $\Gamma(t) > 0$ and $\dot{\Gamma}(t) \leq 0$; (2) $\lim_{t \to T_j} \Gamma(t) = \Gamma_{jT}$; (3) $\Gamma(t) = \Gamma_{jT}$ for $\forall t \geq T_j$ with $T_j$ being the predefined fixed time after which the boundary ceases to contracting.*

$$\Gamma_{jv} = \begin{cases} \Gamma_{jv0}\tanh\left(\frac{\lambda_j t}{t - T_{jv}}\right) + \Gamma_{jv0} + \Gamma_{jvT}, t \in [0, T_{jv}) \\ \Gamma_{jvT}, \qquad\qquad\qquad\qquad\qquad t \in [T_{jv}, \infty) \end{cases} \quad (6)$$

*where $\Gamma_{jv0}$, $\Gamma_{jvT}$ and $\Gamma_{jvT}$ are the initial and final boundary values, $\lambda_j$ is the decay rate, $j = 1, 2$ and $T_{jv}$ is the predefined fixed time after which the boundary ceases to contracting.*

**Definiion 2.** *[16] FTTPF is a function designed to evaluate and ensure the system's tracking performance over a fixed time, dictating how the tracking error should decrease over time to meet specific performance criteria by a predefined deadline.*

$$\varphi_v(t) = \begin{cases} e^{-\frac{k_v t}{T_{fv} - t}}, & t \in [0, T_{fv}) \\ 0, & t \in [T_{fv}, \infty) \end{cases} \quad (7)$$

*Equation (7) is concretely constructed as an FTTPF with the following properties : (1)$\varphi(0) = 1$; (2) $\lim_{t \to T_{fv}} \varphi(t) = 0$ and $\varphi(t) = 0$ for $\forall t \geq T_{fv}$ with $T_{fv}$ being a prescribed settling time. $\Gamma_{jv0}$, $\Gamma_{jvT}$, $\lambda_j$, $T_{jv}$, $T_{fv}$ and $k_v$ are positive constant.*

## B. Funnel Error Transformation

In this paper, by embedding FTTPF $\varphi_v(t)$ we construct a new error $\chi(t)$ variable with 0 initial value as in (9).

$$z_1 = \eta - \eta_d \quad (8)$$

$$\chi(t) = z_1(t) - z_1(0)\varphi_v(t) = \eta - \eta_d - z_1(0)\varphi_v(t) \quad (9)$$

Then, $\Gamma_{jv}$, $j = 1, 2$, is applied to ensure that the following symmetry performance constraints on $\chi(t)$ which are satisfied.

$$-\Gamma_{1v} < \chi(t) < \Gamma_{2v} \quad (10)$$

where $\eta_d = [x_d, \; y_d, \; \psi_d]^\top$ represents the desired position of the ship DP system. Besides, to simplify the design of the controller, $T_{1v} = T_{2v}$ is adopted in this paper. In order to comply with the definition of $\chi(t)$ and the requirements of (9), $\chi(0) = z_1(0) - z_1(0)\varphi_v(0) = 0$ guarantees that the initial state $-\Gamma_{1v}(0) < \chi(0) < \Gamma_{2v}(0) \Leftrightarrow -\Gamma_{1v}(0) + z_1(0) < z_1(0) <$

$\Gamma_{2v}(0) + z_1(0)$ is always satisfied, which implicitly means that $\Gamma_{1v}$ and $\Gamma_{2v}$ no longer need to be redesigned in order to keep the characteristic that initial value is 0 of the new error.

By introducing the constructed $\Gamma_{1v}$ and $\Gamma_{2v}$, the maximum overshoot, settling time, and steady boundaries of $\chi(t)$ can be determined by $\max\{\Gamma_{1v0} + \Gamma_{1vT}, \; \Gamma_{2v0} + \Gamma_{2vT}\}$, $T_{jv}$ and $\Gamma_{jvT}$, respectively. The changing of $z_1(t)$ is required to be preassigned over $[0, T_{fv})$ due to $-\Gamma_{1v}(t) + z_1(0)\varphi_v(t) < z_1(t) < \Gamma_{2v}(t) + z_1(0)\varphi_v(t)$ for $\forall t \in [0, T_{fv})$. From the above analysis, (10) can be reformulated as:

$$-\Gamma_1(t) < \chi(t) = z_1(t) - z_1(0)\Phi_1 < \Gamma_2(t), \forall t \geq 0 \quad (11)$$

where $\Gamma_1 = [\Gamma_{1u}, \; \Gamma_{1v}, \; \Gamma_{1r}]^\top$, $\Gamma_2 = [\Gamma_{2u}, \; \Gamma_{2v}, \; \Gamma_{2r}]^\top$ and $\Phi_1 = [\varphi_u, \; \varphi_v, \; \varphi_r]^\top$.

Although the extant FC results can tuned the transient and steady-state responses of $z_1$, the corresponding problem is the need to rely on specific initial conditions. To solve this problem, inspiring from [17], we introduce the following variable transformation:

$$\vartheta(t) = \chi(t) + \mu(t) \quad (12)$$

with

$$\mu(t) = (\Gamma_1(t) - \Gamma_2(t))/2, \; \omega(t) = (\Gamma_1(t) + \Gamma_2(t))/2 \quad (13)$$

From (12) and (13), (11) is equivalent to

$$-\omega(t) < \vartheta(t) < \omega(t) \quad (14)$$

To improve control performance and achieve control objectives, the funnel error transformation as given by equation (15) is applied.

$$\xi_1(t) = \frac{\vartheta(t)}{\sqrt{\omega^2(t) - \vartheta^2(t)}} \quad (15)$$

The derivation of (15) yields $\dot{\xi}_1$

$$\dot{\xi}_1(t) = \Phi_2\left(\dot{\eta} - \dot{\eta}_d - z_1(0)\dot{\Phi}_1(t) + \dot{\mu}(t) - \vartheta(t)\dot{\omega}(t)/\omega(t)\right) \quad (16)$$

where $\Phi_2 = \omega^2(t)/\sqrt{(\omega^2(t) - \vartheta^2(t))^3} > 0$. It should be noted that for complex representations of $\dot{\xi}_1$, NNs are employed to approximate the uncertain terms. In the subsequent function formulations, function arguments are omitted to simplify the presentation and improve readability.

## IV. ADAPTIVE PT FUNNEL CONTROL DESIGN FOR DYNAMIC POSITIONED SHIPS

In this section, adaptive parameters are introduced using NNs for online approximation of the uncertainty terms arising during the controller design process. The Backstepping means is utilized to design the virtual controller $\alpha_v$ and the control law $\tau$ for the second-order ship motion mathematical model (1) and (2). The DSC technique is applied to address the complexity in deriving $\alpha_v$. The controller design procedure consists of two steps for the attitudes and velocity parts. The specific details of the controller design are detailed in IV-A, and the corresponding stability analysis is detailed in IV-B.

## A. Controller Design

Step 1: In the ship's DP system, the reference attitude signal $\eta_d$ is a constant with derivative 0, meaning $\dot{\eta}_d = 0$. It is noted that in the derivative form of the boundary transformation error $\xi_1$, $\Phi_2(-z_1(0)\Phi_1(t) + \dot{\mu}(t) - \vartheta(t)\dot{\omega}(t)/\omega(t))$ represents the unknown function vector. It can be obtained as (17) by using RBF-NNs $F_1(\eta)$.

$$
\begin{aligned}
F_1(\eta) &= \Phi_2\left(-z_1(0)\Phi_1(t) + \dot{\mu}(t) - \vartheta(t)\dot{\omega}(t)/\omega(t)\right) \\
&= S_1(\eta)A_1\eta + \varepsilon_\eta \\
&= \begin{bmatrix} s_x(\eta) & 0 & 0 \\ 0 & s_y(\eta) & 0 \\ 0 & 0 & s_\psi(\eta) \end{bmatrix} \begin{bmatrix} A_x \\ A_y \\ A_\psi \end{bmatrix} \begin{bmatrix} x \\ y \\ \psi \end{bmatrix} + \begin{bmatrix} \varepsilon_x \\ \varepsilon_y \\ \varepsilon_\psi \end{bmatrix}
\end{aligned}
\tag{17}
$$

where $\varepsilon_\eta$ is corresponding upper bound vector. $s_x(\eta) = s_y(\eta) = s_\psi(\eta)$ due to these RBF functions are with the same input vector $\upsilon$. Let $\theta_1 = \|A_1\eta\|^2$, where $\hat{\theta}_1$ represents the estimated values of $\theta_1$. From the above analysis, the immediate virtual controller $\alpha_\upsilon$ is determined as shown in (18).

$$
\alpha_\upsilon = -\frac{1}{\Phi_2 J(\psi)}\left(k_1\xi_1 + \frac{S_1^T S_1 \hat{\theta}_1}{2a_1^2}\xi_1 + \frac{1}{4}\|\Phi_2\|^2\xi_1\right) \tag{18}
$$

where $k_1$ is a strictly positive diagonal matrix of parameters. the DSC technique, i.e., a first-order low-pass filter (19), is applied here, considering that the derivatives of a are difficult to obtain and complex in form.

$$
t_\upsilon\dot{\beta}_\upsilon + \beta_\upsilon = \alpha_\upsilon, \beta_\upsilon(0) = \alpha_\upsilon(0) \tag{19}
$$

$t_\upsilon$ is a constant time-related matrix, and the input velocity vector signal $\alpha_\upsilon$ is transformed into the output velocity vector $\beta_\upsilon$ which is the reference vector for the velocity signal in the second step. Defining the error vector $q_\upsilon = [q_u, q_\upsilon, q_r]^\top = \alpha_\upsilon - \beta_\upsilon$, $z_2 = \beta_\upsilon - \upsilon$, the derivative of $q_\upsilon$ is acquired along with (18) and (19).

$$
\begin{aligned}
\dot{q}_\upsilon &= -\dot{\beta}_\upsilon + \dot{\alpha}_\upsilon \\
&= t_\upsilon^{-1}q_\upsilon + B_\upsilon\left(z_1, \dot{z}_1, \psi, r, \hat{\theta}_1, \dot{\hat{\theta}}_1\right)
\end{aligned}
\tag{21}
$$

where $B_\upsilon = [B_u(\cdot), B_\upsilon(\cdot), B_r(\cdot)]^\top$ is a vector which includes 3 bounded continuous functions. Otherwise, there are the unknown positive value $\bar{B}_\upsilon = \left[\bar{B}_u(\cdot), \bar{B}_\upsilon(\cdot), \bar{B}_r(\cdot)\right]^\top$ such that $|B_\upsilon| \le \bar{B}_\upsilon$. Then, the dynamic error $z_1$ can be expressed as (21).

Step 2: Together with the time derivative (19) of $z_2$ yields the corresponding result as (22).

$$
\dot{z}_2 = \dot{\beta}_\upsilon - \dot{\upsilon} = M^{-1}\left(M\dot{\beta}_\upsilon + D(\upsilon)\upsilon - \tau - \tau_d\right) \tag{22}
$$

It is noted that $D(\upsilon)\upsilon$ is the uncertain term in the dynamic positioning system. Similar to the treatment of the unknown function vector in the first step, RBF-NNs are used to approximate this uncertainty term as follows:

$$
\begin{aligned}
F_2(\upsilon, A_2) &= S_2(\upsilon)A_2\upsilon + \varepsilon_\upsilon \\
&= \begin{bmatrix} s_u(\upsilon) & 0 & 0 \\ 0 & s_\upsilon(\upsilon) & 0 \\ 0 & 0 & s_r(r) \end{bmatrix} \begin{bmatrix} A_u \\ A_\upsilon \\ A_r \end{bmatrix} \begin{bmatrix} u \\ \upsilon \\ r \end{bmatrix} + \begin{bmatrix} \varepsilon_u \\ \varepsilon_\upsilon \\ \varepsilon_r \end{bmatrix}
\end{aligned}
\tag{23}
$$

In (23), the output vector $F_2 = [f_2(u), \ f_2(\upsilon), \ f_2(r)]$ contains three components correspond to the $u$, $\upsilon$, $r$ component velocities. Let $\theta_2 = \|A_2\upsilon\|^2$, where $\hat{\theta}_2$ represents the estimated values of $\theta_2$. The application of RBF-NNs simplifies the design of subsequent controller and adaptive laws, while reducing the computational complexity of the algorithms to enhance control performance.

In the derivation of formulas involving NNs, three key applications of Youngs inequality are highlighted below:

$$
\begin{aligned}
\Phi_2\xi_1 F_1 &\le \xi_1(S_1 A_1\eta + \varepsilon_\eta) \\
&\le \frac{S_1^T S_1\|A_1\eta\|^2}{2a_1^2}\xi_1^2 + \frac{1}{2}a_1^2 + \xi_1^2 + \frac{1}{4}\varepsilon_\eta^2
\end{aligned}
\tag{24}
$$

$$
\begin{aligned}
z_2 F_2 &\le z_2(S_2 A_2\upsilon + \varepsilon_\upsilon) \\
&\le \frac{S_2^T S_2\|A_2\upsilon\|^2}{2a_2^2}z_2^2 + \frac{1}{2}a_2^2 + z_2^2 + \frac{1}{2}\varepsilon_\upsilon^2
\end{aligned}
\tag{25}
$$

$$
-\Phi_2 J(\psi)\xi_1 q_\upsilon \le \|q_\upsilon\|^2 + \frac{1}{4}\|\Phi_2\|^2\xi_1^2 \tag{26}
$$

Based on the above analysis, (27) is chosen as the control input $\tau$ for the ship dynamic positioning system in this paper. Equation (28), (29) are the expression for the adaptive rate $\dot{\hat{\theta}}_1$, $\dot{\hat{\theta}}_2$.

$$
\tau = k_2 z_2 + \dot{\beta}_\upsilon + \frac{S_2^T S_2 \hat{\theta}_2}{2a_2^2}z_2 + \Phi_2 J(\psi)\xi_1 \tag{27}
$$

$$
\dot{\hat{\theta}}_1 = \frac{\gamma_1 S_1^T S_1 \hat{\theta}_1}{2a_1^2} - \varsigma_1\hat{\theta}_1 \tag{28}
$$

$$
\dot{\hat{\theta}}_2 = \frac{\gamma_2 S_2^T S_2 \hat{\theta}_2}{2a_2^2} - \varsigma_2\hat{\theta}_2 \tag{29}
$$

where $k_2$ is a strictly negative diagonal parameter matrix, $a_1$, $a_2$, $\gamma_1$, $\gamma_2$, $\varsigma_1$ and $\varsigma_2$ is positive design constants. It can be obviously observed that the designed controller has a very simple form, which significantly reduces the computational load and memory usage. Next, the semi-global uniformly ultimately bounded (SGUUB) stability of the DP system is demonstrated after incorporating the proposed algorithm, through a stability analysis.

## B. Stability Analysis

Select the Lyapunov function as following:

$$
V = \frac{1}{2}\xi_1^\top\xi_1 + \frac{1}{2}z_2^\top M z_2 + \frac{1}{2}q_\upsilon^\top q_\upsilon + \frac{1}{2\gamma_1}\tilde{\theta}_1^\top\tilde{\theta}_1 + \frac{1}{2\gamma_2}\tilde{\theta}_1^\top\tilde{\theta}_1 \tag{30}
$$

where $\tilde{\theta}_1 = \hat{\theta}_1 - \theta_1$, and $\tilde{\theta}_2 = \hat{\theta}_2 - \theta_2$. By considering $\vartheta(t)/\sqrt{\omega^2(t) - \vartheta^2(t)}$ and $z_2 = \beta_v - \upsilon$, the time derivative of $V$ is expressed as:

$$\dot{V} = \xi_1^\top \dot{\xi}_1 + z_2{}^T M \dot{z}_2 + q_v{}^T \dot{q}_v + \frac{1}{\gamma_1} \tilde{\theta}_1^T \dot{\hat{\theta}}_1 + \frac{1}{\gamma_1} \tilde{\theta}_2^T \dot{\hat{\theta}}_2 \quad (31)$$

According to (24), (26), $\|J(\psi)\| = 1$ and Young's inequality, it is obtained that

$$\xi_1^\top \dot{\xi}_1 = \xi_1^\top \left[ \Phi_2 J(\psi)(\alpha_v - (z_2 - q_v)) + S_1 A_1 \eta + \varepsilon_\eta \right]$$
$$= \xi_1^\top \left\{ \Phi_2 J(\psi) \left( -\frac{1}{\Phi_2} J(\psi)^{-1} \left( k_1 \xi_1 + \frac{S_1^\top S_1 \hat{\theta}_1}{2a_1{}^2} \xi_1 \right. \right. \right.$$
$$\left. \left. \left. + \frac{1}{4} \|\Phi_2\|^2 \xi_1 \right) - (z_2 - q_v) \right) \right\} + \xi_1^\top S_1 A_1 \eta + \xi_1^\top \varepsilon_\eta$$
$$\leq \xi_1^T \left\{ -k_1 \xi_1 - \frac{S_1^T S_1 \hat{\theta}_1}{2a_1{}^2} \xi_1 - \frac{1}{4} \| \Phi_2 \|^2 \xi_1 \right\}$$
$$- \xi_1^\top \Phi_2 J(\psi) z_2 - \xi_1^\top \Phi_2 J(\psi) q_v + \frac{S_1^\top S_1 \|A_1 \eta\|^2}{2a_1{}^2} \xi_1^\top \xi_1$$
$$+ \frac{1}{2} a_1{}^2 + \xi_1^\top \xi_1 + \frac{1}{4} \varepsilon_\eta{}^2$$
$$\leq -k_1 \xi_1^\top \xi_1 + \frac{S_1^\top S_1 \left( \theta_1 - \hat{\theta}_1 \right)}{2a_1{}^2} \xi_1^\top \xi_1 - \frac{1}{4} \| \Phi_2 \|^2 \xi_1^\top \xi_1$$
$$- \xi_1^\top \Phi_2 J(\psi) z_2 + \| q_v \|^2 + \frac{1}{4} \| \Phi_2 \|^2 \xi_1^\top \xi_1 + \xi_1^\top \xi_1$$
$$+ \frac{1}{4} \varepsilon_\eta{}^2 + \frac{1}{2} a_1{}^2$$
$$\leq -k_1 \xi_1^\top \xi_1 - \frac{S_1^T S_1 \tilde{\theta}_1}{2a_1{}^2} \xi_1^\top \xi_1 - \xi_1^\top \Phi_2 J(\psi) z_2$$
$$+ \| q_v \|^2 + \xi_1^\top \xi_1 + \frac{1}{4} \varepsilon_\eta{}^2 + \frac{1}{2} a_1{}^2 \quad (32)$$

In view of (22), (23), (25) and (27), $\|J(\psi)\| = 1$ and Young's inequality, it follows that

$$z_2^\top M \dot{z}_2 = z_2^\top M \left[ M^{-1} \left( M \dot{\beta}_v + D\upsilon - \tau - \tau_d \right) \right]$$
$$= z_2^\top \left[ M \dot{\beta}_v + F_2 - \left( k_2 z_2 + \dot{\beta}_v + \frac{S_2^T S_2 \hat{\theta}_2}{2a_2{}^2} z_2 \right) \right.$$
$$\left. - \Phi_2 R(\psi) \xi_1 - \tau_d \right]$$
$$\leq z_2^\top (M - I) \dot{\beta}_v + \frac{S_2^\top S_2 \hat{\theta}_2}{2a_2{}^2} z_2^\top z_2 + \frac{1}{2} a_2{}^2 + z_2^\top z_2$$
$$+ \frac{1}{2} \varepsilon_v{}^2 - k_2 z_2^\top z_2 - \frac{S_2^\top S_2 \hat{\theta}_2}{2a_2{}^2} z_2 + \Phi_2 R(\psi) z_2^\top \xi_1 - z_2^\top \tau_d$$
$$\leq z_2^\top (M - I) \dot{\beta}_v + \frac{S_2^\top S_2 \tilde{\theta}_2}{2a_2{}^2} z_2^\top z_2 + \frac{1}{2} a_2{}^2 + z_2^\top z_2$$
$$+ \frac{1}{2} \varepsilon_v{}^2 - k_2 z_2^\top z_2 + \Phi_2 R(\psi) z_2^\top \xi_1 - z_2^\top \tau_d \quad (33)$$

It is worth noticing that

$$z_2 (M - I) \dot{\beta}_v \leq \left\| (M - I) t_v{}^{-1} \right\|_F^2 \|z_2\|^2 + \frac{1}{4} \|q_v\|^2 \quad (34)$$

$$-z_2 \tau_d \leq z_2^\top z_2 + \frac{\tau_d{}^\top \tau_d}{4} \quad (35)$$

Note that $I$ is the identity matrix. Then (33) becomes

$$z_2^\top M \dot{z}_2 \leq \left\| (M - I) t_v{}^{-1} \right\|_F^2 \|z_2\|^2 + 2z_2^\top z_2 - k_2 z_2^\top z_2$$
$$- \frac{S_2^\top S_2 \tilde{\theta}_2}{2a_2{}^2} z_2^\top z_2 + \frac{1}{4} \|q_v\|^2 + \Phi_2 R(\psi) z_2^\top \xi_1$$
$$+ \frac{\tau_d{}^\top \tau_d}{4} + \frac{1}{2} a_2{}^2 + \frac{1}{2} \varepsilon_v{}^2 \quad (36)$$

Incorporating adaptive law (28) and $\tilde{\theta}_2 = \hat{\theta}_2 - \theta_2$, (37) and (38) is obtained.

$$\frac{1}{\gamma_1} \tilde{\theta}_1^\top \dot{\hat{\theta}}_1 \leq \frac{1}{\gamma_1} \tilde{\theta}_1^\top \left( \frac{\gamma_1 S_1^\top S_1 \xi_1^\top \xi_1}{2a_1{}^2} - \varsigma_1 \hat{\theta}_1 \right)$$
$$\leq \frac{S_1^\top S_1 \tilde{\theta}_1^\top \xi_1^\top \xi_1}{2a_1{}^2} - \frac{\varsigma_1 \tilde{\theta}_1^\top \hat{\theta}_1}{\gamma_1} \quad (37)$$

$$\tilde{\theta}_1^\top \hat{\theta}_1 \leq \tilde{\theta}_1^\top (\tilde{\theta}_1 + \theta_1)$$
$$\leq \tilde{\theta}_1^\top \tilde{\theta}_1 + \tilde{\theta}_1^\top \tilde{\theta}_1^\top$$
$$\leq 2\tilde{\theta}_1^\top \tilde{\theta}_1 + \frac{1}{4} \theta_1{}^2 \quad (38)$$

Substituting (38) into (37), one gets

$$\frac{1}{\gamma_1} \tilde{\theta}_1^\top \dot{\hat{\theta}}_1 \leq \frac{S_1^\top S_1 \tilde{\theta}_1^\top \xi_1^\top \xi_1}{2a_1{}^2} - \frac{2\varsigma_1}{\gamma_1} \tilde{\theta}_1^\top \tilde{\theta}_1 - \frac{\varsigma_1}{4\gamma_1} \theta_1{}^2 \quad (39)$$

As the same as above steps, another gets:

$$\frac{1}{\gamma_2} \tilde{\theta}_2^\top \dot{\hat{\theta}}_2 \leq \frac{S_2^\top S_2 \tilde{\theta}_2^\top z_2^\top z_2}{2a_2{}^2} - \frac{2\varsigma_2}{\gamma_2} \tilde{\theta}_2^\top \tilde{\theta}_2 - \frac{\varsigma_2}{4\gamma_2} \theta_2{}^2 \quad (40)$$

Using (21) and Young's inequality, $q_v{}^\top \dot{q}_v$ follows that

$$q_v{}^\top \dot{q}_v \leq - \sum_{i=u,v,r} \left( \frac{q_i{}^2}{t_i} - \frac{B_i^2 q_i \bar{B}_i^2}{2b \bar{B}_i^2} - \frac{b}{2} \right)$$
$$\leq - \sum_{i=u,v,r} \left[ \left( \frac{1}{t_i} - \frac{\bar{B}_i^2}{2b} \right) q_i{}^2 + \left( 1 - \frac{B_i^2}{\bar{B}_i^2} \right) \frac{\bar{B}_i^2 q_i{}^2}{2b} - \frac{b}{2} \right]$$
$$\leq - \sum_{i=u,v,r} \left[ \left( \frac{1}{t_i} - \frac{\bar{B}_i^2}{2b} \right) q_i{}^2 \right] + \frac{3b}{2} \quad (41)$$

Submitting (32) (36) (39) (40) and (41) into (31), the time

derivative $\dot{V}$ is written as (42).

$$\dot{V} \leq -(k_1 - I)\xi_1^\top \xi_1 - (k_2 - 2I)z_2^\top z_2 + \left\| (M-I)t_v^{-1} \right\|_F^2 \|z_2\|^2$$
$$+ \frac{5}{4}\| q_v \|^2 - \sum_{i=u,v,r} \left[ \left( \frac{1}{t_i} - \frac{\bar{B}_i^2}{2b} \right) q_i^2 \right] - \frac{2\varsigma_1}{\gamma_1}\tilde{\theta}_1^\top \tilde{\theta}_1$$
$$- \frac{2\varsigma_2}{\gamma_2}\tilde{\theta}_2^\top \tilde{\theta}_2 - \xi_1^\top \Phi_2 J(\psi)z_2 + \xi_1 \Phi_2 R(\psi)z_2^\top - \frac{S_1^\top S_1 \tilde{\theta}_1}{2a_1^2}\xi_1^\top \xi_1$$
$$+ \frac{S_1^\top S_1 \tilde{\theta}_1^\top \xi_1^\top \xi_1}{2a_1^2} - \frac{S_2^T S_2 \tilde{\theta}_2}{2a_2^2}z_2^\top z_2 + \frac{S_2^T S_2 \tilde{\theta}_2^\top \xi_2^\top \xi_2}{2a_2^2}$$
$$+ \frac{1}{4}\varepsilon_\eta^2 + \frac{1}{2}a_1^2 + \frac{\tau_d^\top \tau_d}{4} + \frac{1}{2}a_2^2 + \frac{1}{2}\varepsilon_v^2 - \frac{\varsigma_1}{4\gamma_1}\theta_1^2$$
$$- \frac{\varsigma_2}{4\gamma_2}\theta_2^2 + \frac{3b}{2}$$
$$\leq -2aV + \varrho \tag{42}$$

where $a = \lambda_{\min}\left\{ -(k_1 - I), -(k_2 - 2I) + \left\| (M-I)t_v^{-1} \right\|_F, \right.$

$\left. \left\{ 5/4 - \sum_{i=u,v,r}\left[ 1/t_i - \bar{B}_i/2b \right] \right\}, -2\varsigma_1/\gamma_2 - 2\varsigma_2/\gamma_2 \right\}, \varrho = 1/(4\varepsilon_\eta^2)$
$+ 1/(2a_1^2) + \tau_d^\top \tau_d/4 + 1/(2a_2^2) + 1/(2\varepsilon_v^2) - \varsigma_1\theta_1^2/(4\gamma_1)$
$- \varsigma_2\theta_2^2/(4\gamma_2) + 3b/2.$

By integrating both sides of equation (42), we obtain:

$$V(t) \leq \left( V(0) - \frac{\varrho}{2a} \right)e^{(-2at)} + \frac{\varrho}{2a} \tag{43}$$

According to the closed-loop gain shaping algorithm [18], all errors variables in closed-loop system decrease to the compact set $\Omega := \left\{ \left( \xi_1, z_2, q_v, \tilde{\theta}_1, \tilde{\theta}_2 \right) \middle| \|\xi_1\| \leq C_0, C_0 > \sqrt{\varrho/a} \right\}$ as $t \to \infty$ by choosing appropriate parameters. $C_0 > \sqrt{\varrho/a}$ is a positive constant. Thus, the closed-loop control system is SGUUB stable under the proposed control scheme, given the positive constant $C_0$, where all signal errors in the closed-loop system can be made arbitrarily small.

## V. SIMULATION

In this section, to verify the effectiveness of the proposed prescribed-time algorithm, a simulation example for a supply ship (length: 76.2m, mass: $4.591 \times 10^6$kg) equipped with a DP system is executed and compared with the Optimum-seeking Guidance scheme (OSG) in [19] and robust control scheme in [20]. The ship mathematic model parameters are presented in TABLE I. In the modeling of ship Dynamic Positioning (DP) systems, it is essential to precisely characterize and predict the impacts of environmental disturbances, such as wind, waves, and ocean currents on the ship's performance. For the sake of simplifying the ship model and facilitating the design and testing of control algorithms, these environmental disturbances are approximated and modeled using a sine-cosine function (44).

TABLE I
MODEL PARAMETERS

| Indexes | Values | Indexes | Values |
|---|---|---|---|
| $X_{\dot{u}}$ | $-0.72 \times 10^6$ | $X_{\dot{u}}$ | $5.0242 \times 10^4$ |
| $Y_{\dot{v}}$ | $-3.6921 \times 10^6$ | $Y_v$ | $2.7229 \times 10^6$ |
| $Y_{\dot{r}}$ | $-1.0234 \times 10^6$ | $Y_r$ | $-4.3933 \times 10^6$ |
| $I_z - N_{\dot{r}}$ | $3.7454 \times 10^9$ | $Y_{|v|v}$ | $1.7860 \times 10^4$ |
| $X_{|u|u}$ | $1.0179 \times 10^3$ | $Y_{|v|r}$ | $-3.0068 \times 10^5$ |
| $N_v$ | $-4.3821 \times 10^6$ | $N_r$ | $4.1894 \times 10^6$ |
| $N_{|v|v}$ | $-2.4684 \times 10^5$ | $N_{|v|r}$ | $6.5759 \times 10^6$ |

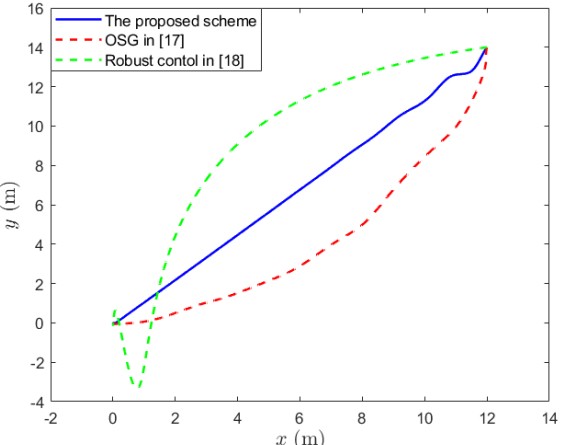

Fig. 1.   Trajectory of the ship in $xy$-plane.

$$\begin{cases} \tau_{du} = 2\left(1 + 35\sin\left(0.2t + 15\cos(0.5t)\right)\right)(\text{N}) \\ \tau_{dv} = 2\left(1 + 30\cos\left(0.4t + 20\cos(0.1t)\right)\right)(\text{N}) \\ \tau_{dr} = 3\left(1 + 30\cos\left(0.3t + 10\sin(0.5t)\right)\right)(\text{N} \cdot \text{m}) \end{cases} \tag{44}$$

$k_1 = \text{diag}\left[0.2, \ 0.38, \ 0.20\right], \ k_2 = \text{diag}\left[44, \ 12.8, \ 78.1\right];$
$t_v = 0.05 \times I; a_1 = a_2 = 80; \ \gamma_1 = \gamma_2 = 0.5; \ \varsigma_1 = \varsigma_2 = 0.5;$
$T_{jv} = \left[T_{ju}, \ T_{jv}, \ T_{jr}\right] = \left[80s, \ 80s, \ 90s\right];$
$T_{fv} = \left[T_{fu}, \ T_{fv}, \ T_{fr}\right] = \left[80s, \ 80s, \ 90s\right]; \tag{45}$

In this simulation, the desired attitude is set to $\eta_d = [0\text{m}, \quad 0\text{m}, \quad 0\text{deg}]$. The initial states are set to $\eta(0) = [12\text{m}, \quad 14\text{m}, \quad 10\text{deg}], v(0) = [0\text{m/s}, \quad 14\text{m/s}, \quad 10\text{deg/s}]$. The concrete parameters values setting follows (45). Besides, the RBF-NNs for $F_1$ and $F_1$ consisted of 25 nodes with centers spaced in $[-2.5\text{m/s}, 2.5\text{m/s}]$ for $x$, $y$, $u$ and $r$, $[-0.16 \text{ m/s}, \ 0.16 \text{ m/s}]$ for $\psi$ and $r$, respectively. For the comparison algorithms, corresponding parameters refers to [19] and [20].

Fig. 1 exhibits simulation results under the proposed algorithm, OSG and robust control making the ship stay at the desired attitude in the $xy$-plant. It is clear that the proposed algorithm provides a more satisfactory trajectory accuracy compared to the algorithms for comparison. Fig. 2 illustrates that the ship attitude $x$, $y$ and $\psi$ are stabilized to the desired attitude near the prescribed-time $T_{jv}$. The proposed scheme achieves faster stabilization compared to the schemes for comparison. The velocities of surge, sway and yaw are showed

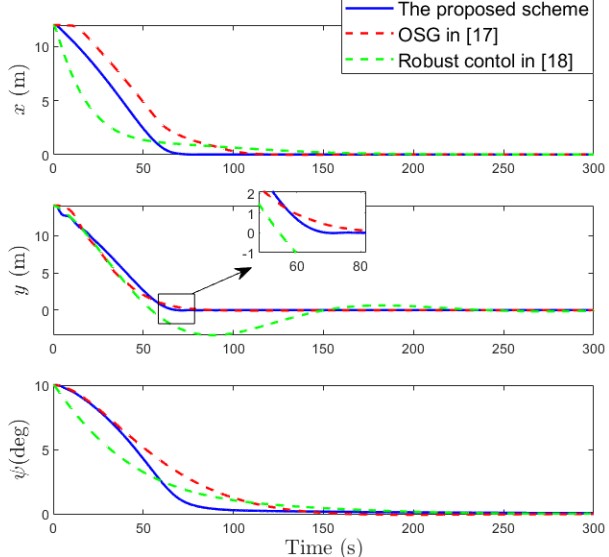

Fig. 2. Ship's actual position $(x, y)$ and heading $\psi$.

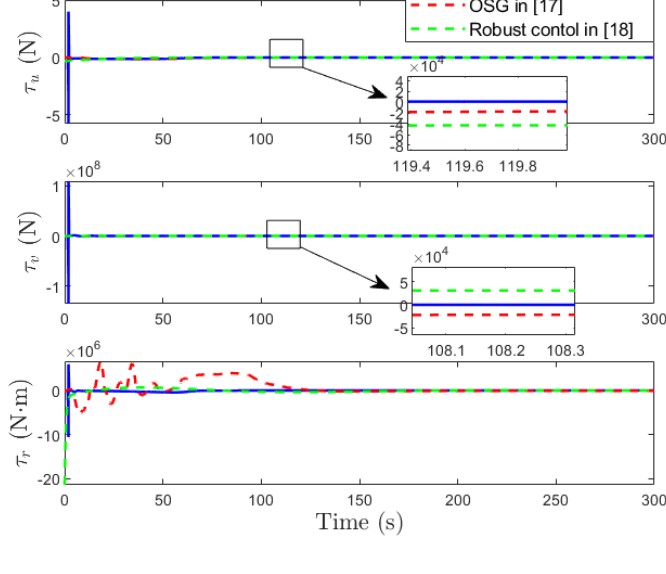

Fig. 4. Ship's surge force $\tau_u$, sway force $\tau_v$ and yaw force $\tau_r$.

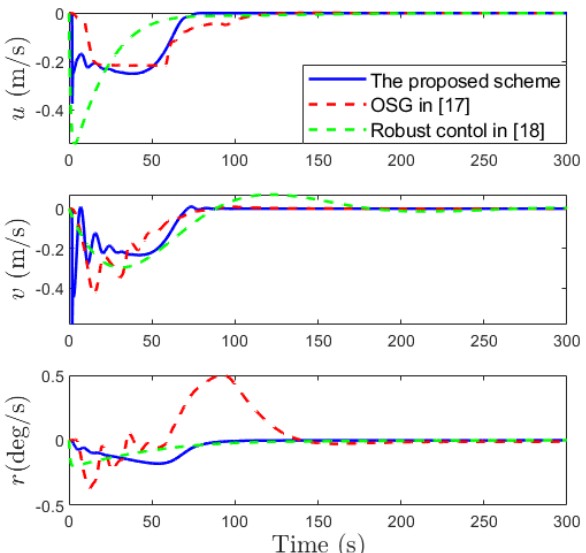

Fig. 3. Ship's surge velocity $u$, sway velocity $v$ and yaw rate $r$.

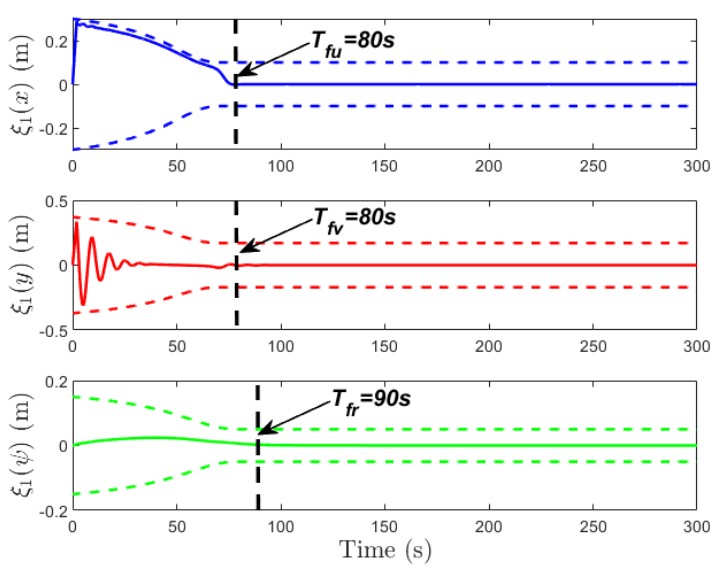

Fig. 5. The new error $\xi_1$ for the simulation with the proposed scheme.

in Fig. 3. The proposed scheme has a improved convergence performance. Fig. 4 illustrates the fluctuation of the values of the three input signals over time. It is apparent that prior to system stabilization, the proposed scheme exhibits superior convergence performance of $\tau_r$ compared to other schemes. Once the system has stabilized, the values of $\tau_u$ and $\tau_v$ in the proposed scheme converge more rapidly towards zero, further outperforming the other schemes in convergence efficiency. Finally, it can be seen that the constructed new error is successfully confined within the boundaries and converges stably to 0 at the moment of settling time $T_{fu}$, $T_{fv}$, $T_{fr}$ as shown in Fig. 5. Additionally, Fig. 6 and Fig. 7 illustrate the fitting performance between the estimated and true values of

the adaptive parameters $\theta_1$ and $\theta_2$, respectively, representing the approximation capability of the RBF-NNs for the system uncertainties terms. It can be observed that, within the permissible margin of error, the RBF-NNs successfully approximate uncertainties terms described by (13) and (17).

In summary, the NNs-based prescribed-time control scheme proposed in this paper demonstrates superior performance and robustness compared to the schemes for comparison. By introducing FTFBs, FTTFBs and constructing new error functions, the control laws are made more concise. Finally, the proposed scheme is validated through simulations to demonstrate its effectiveness on DP ships.

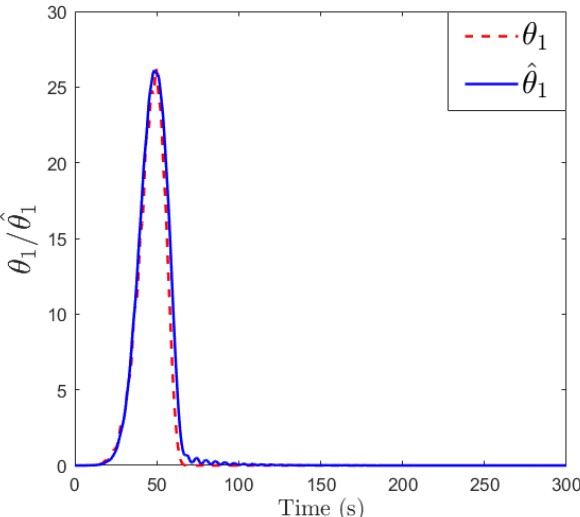

Fig. 6. The estimation performance of $\theta_1$.

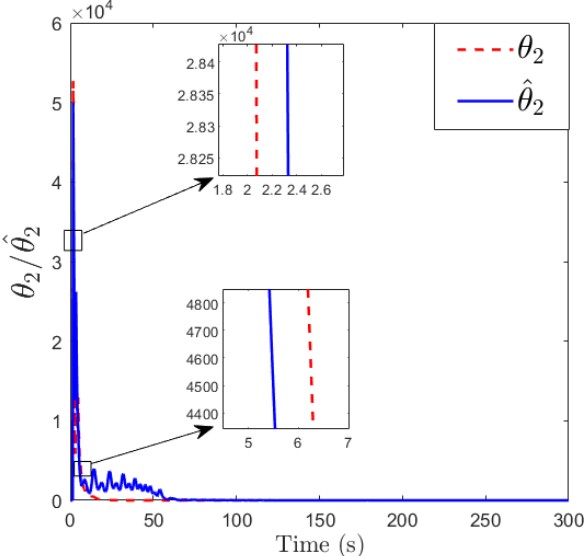

Fig. 7. The estimation performance of $\theta_2$.

## VI. CONCLUTION

In this paper, a novel NNs-based control scheme is proposed for the ship DP system under model uncertain and unknown environmental disturbances, making the new dynamic errors converging within fixed boundaries. The prescribed-time performance of the algorithm is validated through by a simulation example and two comparative simulations with satisfactory results. Consequently, the prescribed-time control algorithm proposed in this paper can be applied to ships performing DP tasks, enabling the ship's dynamic system to achieve more precise time-based prescribed performance.

Given the presence of multiple dynamic actuators in engineering practices related to marine equipment, future research on the proposed algorithm could focus on the issue of actuator control allocation. In addition, the integration of event-triggered control, fault-tolerant control, and blind zone constraints could further enhance the development of this control algorithm toward more advanced and precise control techniques.

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
