# OpenReview forum: "Adaptive Prescribed-time control of Dynamic Positioning ships based on Neural networks"
_IEEE.org/ICIST/2024/Conference — IEEE ICIST 2024 Conference Submission_

### Official Review · Reviewer_KALy · 2024-08-24
**In this paper, a novel NNs-based control method is proposed for the ship DP system under model uncertain and unknown environmental disturbances. This is a solid conference paper.  But there are some issues are made as follows**

**Rating:** 7
**Confidence:** 4

**Review:**

(1) Please unify the nouns, such as ' marine and sea ',' ship and vessel '.
(2) There are some grammar errors in the paper.
(3) There are many formatting issues, such as indentation of the first line.
(4)The conclusion section is too short.

---

### Official Review · Reviewer_37HX · 2024-08-25
**minor repair**

**Rating:** 8
**Confidence:** 3

**Review:**

1. There are citation irregularities in this manuscript, for example: (i) If Eq.1 and Eq. 2 are not first introduced by the authors, it is recommended that the corresponding references be added.
2. Authors should summarize the contributions in the Introduction Part introducing comparison with pervious researches.
3. An analysis of the learning effect of the network on uncertainty should be added to the simulation.

---

### Official Review · Reviewer_9Lja · 2024-08-29
**Accept**

**Rating:** 9
**Confidence:** 5

**Review:**

This paper focuses on designing a controller with a prescribed time function. Overall, the paper presents a novel study that using the designed controller can converge the new error stably for the DP system under modelling uncertain and unknown environmental disturbances. However, several aspects could benefit from clarification, revision, or additional analysis.
1. The author should consider additional simulations or experiments to strengthen conclusions.
2. Some figures could be improved by reorganizing data or adding labels.
3. The discussion of limitations and future work could be expanded to more fields.
4. The language of this paper could be polished.

---

### Decision · Program_Chairs · 2024-09-06

Accept (Oral)